# Effect of Hydrothermal Aging on Damping Properties in Sisal Mat-Reinforced Polyester Composites

**DOI:** 10.3390/polym16020166

**Published:** 2024-01-05

**Authors:** Moisés F. E. Silva, Beatriz R. Silva, Adrielle N. Marques, Silvana Mattedi, Ricardo F. Carvalho

**Affiliations:** 1Construction and Structures Department, Federal University of Bahia, Salvador 40210-630, BA, Brazil; moises.silva@fieb.org.br (M.F.E.S.); reismbea@gmail.com (B.R.S.); adrielle_cte@hotmail.com (A.N.M.); 2SENAI Institute of Technology in Civil Construction, SENAI CIMATEC, Salvador 41650-010, BA, Brazil; 3Chemical Engineering Graduate Program, Federal University of Bahia, Salvador 40210-630, BA, Brazil

**Keywords:** sisal fiber, reinforced polyester composite, hydrothermal aging

## Abstract

Hydrothermal aging is a matter of considerable concern for natural fiber-reinforced polymers; it can alter dimensional stability and induce microcracks and macro strain on the composite structure. This study applied a sorption kinetic model and examined the effects of water on the damping factor of sisal mat-reinforced polyester composites. The experimental data were fitted well using a Boltzmann sigmoid function, suggesting a promising first step toward kinetic water sorption modeling. Additionally, a damping test was carried out using the impulse excitation technique, highlighting the composite material’s dynamic response under varying water absorption conditions. The result showed that damping exhibited sensitivity to water absorption, increasing significantly during the first 24 h of immersion in water, then remained steady over time, inferring a critical time interval. An empirical model proved satisfactory with the correlation coefficient for sorption rates and damping of sisal mat polymeric composites.

## 1. Introduction

Environmental awareness has become increasingly present in current research on developing eco-friendly materials, especially those that meet industrial requirements and stem from low-cost production processes using bio-based materials. Using natural fibers as reinforcement in polymer composites has become more common over the last decade because of several factors, such as their low cost, biodegradability, good mechanical and microstructural properties, low weight, and non-abrasiveness [1]. A good balance between natural fiber damping and stiffness-to-weight ratio is crucial to obtain optimal material performance for applications in different fields, particularly sports equipment, musical instruments, panels, boxes, and other types of supporting and packaging objects [2].

However, the durability and mechanical properties of natural fiber-reinforced polymer composites (NFRCs) when exposed to hydrothermal environment cycles (hot–wet) are still main concerns in industrial applications. David-West highlighted that the stiffness of flax/hemp-hybrid composites decreased with increasing hydrothermal aging, while the quantity of water absorbed impacted energy dissipation [3]. Chilali et al. investigated the effect of water aging on the load–unload cycle of flax fiber-reinforced thermoplastic and thermosetting composites [4]. The authors concluded that load–unload tensile tests accentuate the stiffness loss with aging, due to the microstructural damage and the plasticization phenomena caused by water absorption.

Damping analysis is a quick and nondestructive method for testing materials and has been increasingly used to evaluate the quality of composite interfaces. The interface dissipates part of the energy, which can be measured by vibration damping and the level of adhesion within the fiber/matrix. There are different sources of energy dissipation in fiber-reinforced composites: viscoelastic properties of the matrix and/or fiber materials, interface damping, damping caused by damage, viscoplastic damping, and thermoelastic damping [5]. Moreover, other complementary variables, such as moisture, temperature variations, fiber orientation, fiber volume fraction, and matrix type, contribute to the damping–energy relationship [6]. Berges et al. reported that the plasticization effects due to water molecules absorbed by fibers and the matrix increase the damping of flax fiber-reinforced polymer composites [7]. Cheour et al. highlighted that damping tests could contribute to the evaluation of the damage caused by fatigue fractures owing to excessive and repeated vibrational load on composite materials [8]. Senthilkumar et al. investigated temperature curing of polyester composite-reinforced sisal fibers and its influence on the free vibration properties [9]. The authors pointed out that increasing the curing temperature caused a smaller energy dissipation mechanism, thereby decreasing the damping factor.

Water can also penetrate the free area between the fiber and matrix, increasing the friction loss within them [10]. At the same time, fiber swelling caused by moisture leads to significant changes in the dimensional stability of the composite, inducing cracking in the matrix and damage to the fiber/matrix interface, thereby increasing the energy dissipation of the composites [8]. In contrast, Venkataravanappa et al. reported negligible alterations in dimensional stability after water absorption in hybrid specimens of hemp and banana fibers combined with polyester resin [11]. Their study revealed minimal variation in hybrid composites, whereas significant effects were observed in specimens containing hemp or banana fibers. On the other hand, Duc et al. used vibration beam testing to determine that the loss factor through intra-yarn friction was dominant, leading to energy dissipation [12]. Tang and Yan concluded that it is a challenge to associate the viscoelasticity of the polymer matrix as the most significant contributor to the energy dissipation behavior of fiber-reinforced composites [13].

In this context, to potentialize the use of these bio-based and renewable materials and contribute to attenuating vibrations to a desirable level for structural and non-structural applications, this work aims to investigate the influence on the damping performance of sisal mat-reinforced polyester composites when exposed to different hydrothermal aging.

## 2. Materials and Methods

### 2.1. Materials

The matrix used was a thixotropic, accelerated, and low-viscosity resin of an unsaturated polyester type AROPOL L 50500 T-10 with a viscosity (25 °C) of 250350 cPs and a density of 1.10 g/cm^3^. Hamilton Rios Ltd. (Conceição do Coité, Brazil) supplied the sisal mat used as reinforcement. The weight of the sisal mat was 781.25 g/m^2^, and the physical, mechanical, and chemical properties from the literature are reported in Appendix A. Pre- and post-processing images are exemplified in Figure 1a,b.

### 2.2. Compounding and Processing

Two layers of sisal mats were used as reinforcements for the polyester composites (C2SM). The compounding process was carried out using hot compression molding with a thermohydraulic platen press under 5 tons (0.78 MPa) at 90 °C for 90 min, resulting in 32 × 20 × 0.4 cm^3^ rectangular sheets.

### 2.3. Hydrothermal Aging

Water sorption tests were performed following the ASTM D570 [14]. Four composite specimens (76 × 25 × 4 mm^3^) were dried in an oven at 60 °C until the surface water was removed and the weight loss was stabilized. Next, they were immersed in distilled water at different temperatures (30, 60, and 90 °C) until saturation. Gravimetric analysis measured weight gain (%) compared to the initial dry sample weight. The interval at which the samples were weighed was 2 h.

The moisture equilibrium (or saturation point) was assumed to be the point at which the weight gain was less than 0.001 g. The percentage of water sorption in the composites was calculated using Equation (1):(1)$$M\left(\%\right)=\frac{M_{w}-M_{s}}{M_{s}}\times 100$$
where *M_s_* (g) is the initial weight and *M_w_* (g) is the weight at time *t* (s).

The diffusion behavior in glassy polymers is associated with the penetrant mobility between polymer chains and can be expressed theoretically by the shape of the sorption curve as Fickian behavior, relaxation-controlled, or non-Fickian diffusion, as represented by Equation (2). In this case, a logarithmic model was applied to investigate the linear correlation between parameters *n* and *k*. Equation (3) is based on Equation (2) and allows for the proper definition of different types of diffusion behavior based on the shape of the curve [15]. The values of parameter *n* indicate the type of diffusion: for Fickian behavior, *n* = 0.5; for relaxation, *n* < 0.5; and for non-Fickian behavior, 0.5 < *n* < 1 [16]. The parameter *k* can be used to determine the water affinity of the composites.
(2)$$\frac{M_{\left(t\right)}}{M_{\infty }}=k\cdot t^{n}$$
(3)$$\log\left(\frac{M_{\left(t\right)}}{M_{\infty }}\right)=\log\left(k\right)+n\cdot log(t)$$
where *M*_∞_ (%) is the equilibrium moisture, *M_t_* (%) is the moisture uptake at time *t* (s), and *n* and *k* are nondimensional parameters.

The diffusion coefficient (*D*) was obtained from the slope of the mass uptake and can be expressed by Equation (4):(4)$$D=\pi {\left(\frac{kh}{{4M}_{\infty }}\right)}^{2}$$
where *k* is the initial slope, *M*_∞_ is the maximum weight gain, and *h* is the thickness of the composites.

To compare with Fick’s theory and obtain a sigmoidal fitting correlation for the experimental data, the model based on the Boltzmann equation was applied to predict mass transfer phenomena and can be expressed by Equation (5) below.
(5)$$y=\frac{A_{1}-A_{2}}{1+e^{(x-x_{0})/dx}}+A_{2}$$
where *A*_1_ is considered the initial value of *y* (−∞), *A*_2_ is the final value of *y* (+∞) associated with equilibrium sorption, *x*_0_ is the central inflection point, and dx is the variation in time.

### 2.4. Damping Analysis

The damping factor was evaluated using an impulse excitation technique with a logarithmic decrement. The data acquisition system used Sonelastic^®^ from *ATCP Engenharia Física* for the nondestructive characterization of damping materials following ASTM E756-05 [17]. Six specimens with dimensions of 150 × 25 × 4 mm^3^ were subjected to damping tests. To validate the measurements, ten measurements were performed for each sample under dry and wet conditions. The time intervals were established as 0 (dry) 3, 5, 7, 24, and 48 h. The water bath temperatures were set at 30, 60, and 90 °C. The standard test criteria from the software manual were adopted with frequencies varying from 0.5 to 20 kHz and a maximum acquisition time of 0.683 s.

From the theoretical curve shown in Figure 2, parameters can be obtained, such as the loss factor (tan φ) and the damping factor (ζ), which are interrelated.

### 2.5. Scanning Electron Microscopy (SEM)

To evaluate the damage to the fiber/matrix interface, the composites were frozen by immersion in liquid nitrogen and then cut into cross-sectional pieces. The materials were then coated with gold using a metallizer and imaged using a scanning electron microscope (SEM) (Tescan Mark, São Paulo, Brazil, model LMU-Vega 3, 10 kV, secondary electron detector with working distances between 8 and 25 mm).

## 3. Results and Discussion

### 3.1. Water Sorption Kinetics

The effects of hydrothermal aging on sisal mat composites are presented in Figure 3, in which water uptake (%) is a function of time (s)^0.5^. The first assumption considered experimental compatibility with the Fickian model.

Figure 3, which shows the effect of water on sisal mat composites, reveals a two-stage diffusion process, comprising an initial linear stage followed by an exponential curve approaching saturation. A slight slope in the curve was observed for hydrothermal aging at 60 °C and 90 °C. Increasing the temperature decreased the time necessary to reach equilibrium moisture, as seen in Table 1.

When the temperature was low (e.g., 30 °C), 11 d were required for the composites to become saturated, whereas by raising the temperature to 60 °C or 90 °C, the time to reach the equilibrium state was reduced significantly; thus, saturation was almost tenfold faster. In addition, for sisal mat composites, the dependence of the diffusion coefficient on temperature is noteworthy and corroborates previous studies on natural fiber-reinforced composites [18]. Concerning the diffusion coefficient at room temperature, the results were similar to those obtained by Kumari et al. for sisal polyester composites fabricated by hot compression molding [19]. The authors found a value of 1.65 × 10^−6^ mm^2^/s, while the experimental data showed 6.01 · 10^−6^ ± 5 and 19 · 10^−7^ mm^2^/s. Mazur and Kuziel [20] reported that the hydrophilic nature of lignocellulosic fibers significantly contributes to the increase in absorption rates and the diffusion coefficients in composites.

A drop in the relative water uptake was observed in the composites subjected to extreme hydrothermal aging. According to authors [18,21], composite weight loss can be associated with the low molecular weight of soluble materials, such as resin or lignocellulosic substances, from the fibers. Over time, and by gradually increasing the temperature, particulates drop from the composites to the water. A deleterious and irreversible effect was observed on the sisal fiber and polyester, thus changing the shape of the sorption curve. Yorseng et al. found that microcracks and fiber degradation facilitated new pathways for water infiltration, leading to increased water absorption [22].

In addition, stiffness and strength properties can be strongly influenced by the plasticizing effect of water on the matrix, as suggested by Scida et al. [15]. The higher moisture rate at 90 °C might be associated with a capillary transport mechanism, which becomes more active through microcracks [21]. Accordingly, the hydrophilic nature of the natural fibers affected the water sorption rate, as supported by other studies [15,18].

Figure 4 and Table 2 show the graph and values of the regression model used to distinguish the diffusion cases theoretically. The constants were obtained for hydrothermally aged composites at 30 °C, 60 °C, and 90 °C, and the results were well adjusted for the three temperature conditions, as indicated by R^2^ > 0.9.

It can be observed in Table 2 that all values of *n* obtained from the slope of the log plot (*M_t_*/*M*_∞_) were slightly higher than 0.5, suggesting that the Fickian analytical model did not suitably represent part of the diffusion mechanism and shape curves. Therefore, non-Fickian or anomalous behavior can be assumed for both temperatures, which may indicate that the swelling penetrant stressed the polymer chains and led to a relaxation process.

According to Panthapulakkal and Sain, if the water affinity of the composite increases, so do the values of k [18]. The parameter can be calculated from the intercept of the log (*M_t_*/*M*_∞_) and can be influenced by temperature variation and moisture interaction [15,18]. Table 2 shows the values obtained from k that increased with increasing hydrothermal aging, especially at 90 °C, which had the highest value. Sreekumar et al. also found a deviation from the Fickian curve, with values of *n* varying between 0.2 and 0.3, in particular, a decreasing behavior at highertemperature aging (90 °C) [23]. The authors’ findings support that water absorption under extreme conditions can be associated with the relaxation mechanism, which contributes to the distortion of the polymer network, leading to large-scale segmental motion and further water absorption. Thus, the changes in the relative weight observed in the composites can be attributed to the change in the curve profile and deviation from Fickian diffusion.

In addition, typical Fickian diffusion, which mainly occurs during long-term exposure to hydrothermal conditions, is not appropriate for describing the entire moisture diffusion process [24]. Moreover, the shape of the sorption curve was strongly controlled or driven by molecular diffusion inside the composite materials. Although Fick’s model is commonly used to predict water migration in composites, it needs more precision over the long term in water immersion. As a result, there is a need to develop new models to improve accuracy [25]. Therefore, the application of the Boltzmann model is justified by the observed curve features, which can be attributed to the polymer relaxation process or anomalous behavior. First, the Boltzmann sigmoidal above the midpoint (*x*_0_) rises linearly and is proportional to the concentration gradient, the same as that proposed by the Fickian model. Second, the tops of the curves are in the steady phase, at which the equilibrium plateau can be satisfactorily defined (*A*_2_). Vilaseca et al. also proposed the Boltzmann model for sorption kinetics on starch-based composites reinforced with 10%, 20%, and 30% jute strands and found an exceptionally reliable correlation coefficient [26].

In this study, the Boltzmann model showed good fits for all environmental conditions, as the correlation coefficients, R^2^, were all above 0.98. The accuracy of the sorption kinetics is associated with lower temperatures; at higher temperatures, irreversible damage occurs, which is reflected by a slight reduction in the correlation coefficient (R^2^). The results of the model implementation are shown in Figure 5.

Table 3 lists the parameters of the nonlinear Boltzmann sigmoid model after using it to fit (until converged) the experimental data. The model is in good agreement and is similar to the result obtained from the Fickian diffusion theory. A similarity between *A*_2_ (Boltzmann) and *M*_∞_ (Fickian) suggests an effective way to obtain the parameters of the equilibrium moisture or saturation level of composites, because the *A*_2_ parameter approaches an infinite asymptote, is parallel to the *x*-axis, and is closer to *M*_∞_. Gudayu et al. [27] also found a sigmoidal water absorption curve in both treated and untreated sisal fibers, reaching an equilibrium over time.

### 3.2. Damping Analysis

The curves obtained from the data acquisition system for sisal mat composites, such as the reference state (dry samples), and under different hydrothermal conditions (for 24 and 48 h at 30 °C, 60 °C, and 90 °C) were plotted in terms of amplitude (Vpp) versus time (s). Although the acquisition time was set at 0.683 s, the vibration response stabilized the signal after a short period, as shown in Figure 6.

The signal obtained from the impulse response after water immersion for both hydrothermal conditions was significantly reduced, which can be observed through the initial and final amplitude ratios and shortened oscillation time. The damping, related to the oscillations in the system, decays after a disturbance, which, in this case, is an intense and short excitation impulse. A considerable signal decay was observed in the immersed composites compared to that in the dry state. However, between 24 and 48 h, almost no significant variation was observed for both hydrothermal conditions, thus establishing a critical period. Sample swelling was also evaluated, and the thickness increased from 5% to 8% in the first 24 h of water immersion, suggesting internal tension in the fiber/matrix and a greater friction mechanism between them. Table 4 presents the values of the damping and loss factors at different time intervals and temperatures.

As the damping and loss factors are directly proportional, a considerable increase in these parameters was observed after the increase in temperature variation and the water immersion time. A comparison of reference samples with samples from 24 h and 48 h revealed that the vibration attenuation capacity of the composites was significantly altered. In terms of results, it was found that damping and loss factors increased in the range of 200% to 700%. Cheour et al. reported an increase of 160% in the loss factor of quasi-unidirectional flax fiber-reinforced epoxy composites after saturation, which can be partially associated with the energy dissipation related to the friction process between fibers [8]. Therefore, an SEM image was taken (Figure 6) to evaluate the interface after longer hydrothermal aging. Duc et al. also found 200% higher values of the loss factor of unidirectional flax-fiber composites than synthetic fibers [12]. The friction between the fibers inside the yarn and between yarns was dominant over the damping properties.

Thus, the experimental results showed that temperatures, especially in the 60 °C and 90 °C range, had a more significant effect on the internal composite damage.

Micrographs are shown in Figure 7; as indicated in the SEM images, when the composites were immersed at higher temperatures, fiber/matrix debonding increased proportionally over time. Several phenomena can occur concomitantly. First, due to the low chemical compatibility of sisal fibers with polyester (a nonpolar structure), weak interface adhesion occurs, causing a reduction in the active sites for the initial friction mechanisms. Then, water diffusion through the fiber/matrix interface can act as a continuous lubricant film that occupies voids or zones of less energy. Inside the fibers, water penetrates via microcapillary transition chains within the cell wall, thus causing swelling, which alters the dimensional stability [28]. Kandola et al. report that capillary water can be absorbed if the fiber structure is swollen, and the relative humidity is close to 100% [29].

Figure 7 shows that this hypothesis can be supported at 24 h immersion time; as fibers begin to swell, internal tension is brought into the matrix, which may intensify the friction mechanism and damping properties.

This preliminary result demonstrated the need to accurately evaluate hydrothermal aging on the damping factor at short time intervals within 24 h. For this reason, the time intervals adopted were 3, 5, 7, and 24 h, as shown in Figure 8.

The damping and loss factors also increased as the mat-reinforced composites’ moisture rate (%) increased. Between the time intervals of 3 and 24 h, it was observed that damping and tangent (ϕ) increase proportionally, as water absorption increases. Water sorption has a higher damping capacity and is related to the energy dissipation inside the composites [7], which can be induced by different factors such as matrix viscoelastic behavior, interphase damping, or damping related to internal damage [8].

Figure 9 and Table 5 show the linear correlation model for 0, 3, 5, 7, 24, and 48 h at different aging intervals, and linear equations were used to predict water sorption as a function of the damping properties.

Figure 9 illustrates that the data showed good linear behavior at lower temperatures (30 °C and 60 °C), but that the data changed considerably for hydrothermal conditions at 90 °C. For 30 °C and 60 °C hydrothermal aging, the squared correlation coefficient (R^2^) was satisfactory at 0.96 and 0.97, respectively. At 90 °C, a good correlation could not be established (0.78) and an intensive process of matrix phase degradation occurred. Several reasons for the degradation can be found in the literature, such as plasticization [6], swelling [21], or relaxation behavior [12], which is associated with chain reorientation in the crystalline phase of the matrix. Other authors reported that the energy dissipation is restricted to the matrix and interface region, considering that increased resistance of the interface resulted in lower energy dissipation [30]. The interface contact area is constantly susceptible to friction, and, in this case, the internal friction at 90 °C could have been limited to a shorter interval, partially reducing the energy dissipation after the total process of debonding the fibers and polymers.

Moreover, these results can be corroborated with other empirical models that correlate water sorption with the thickness swelling mechanism of composite materials [31].

## 4. Conclusions

Implementing the Boltzmann sigmoid model for interparticle diffusion was satisfactory in representing the sorption kinetics of sisal composites. In addition, parameter A_2_ from the Boltzmann equation can be compared to *M*_∞_ of the Fickian model, reaching a plateau or equilibrium of moisture. The dynamic behavior of the composites was sensitive in the first hours of immersion (24 h), considered a critical time, with a significant increase in the damping factor and the loss factor, followed by a low variation for a 48 h immersion time, which can be explained as the result of moisture penetration inside the composites, thus leading to matrix degradation, which increased the energy dissipation. At the interface, water penetration in the accessible areas between the composite components contributed to friction loss, which is justified by the values of the loss tangent and damping factor. An empirical linear regression model used to compare water sorption to the damping factor was consistent at lower temperatures; at higher temperatures (90 °C), irreversible matrix and fiber damage was found. The findings from this research highlight the importance of using sisal fibers as elements to dissipate vibrational energy.

## Figures and Tables

**Figure 1 polymers-16-00166-f001:**
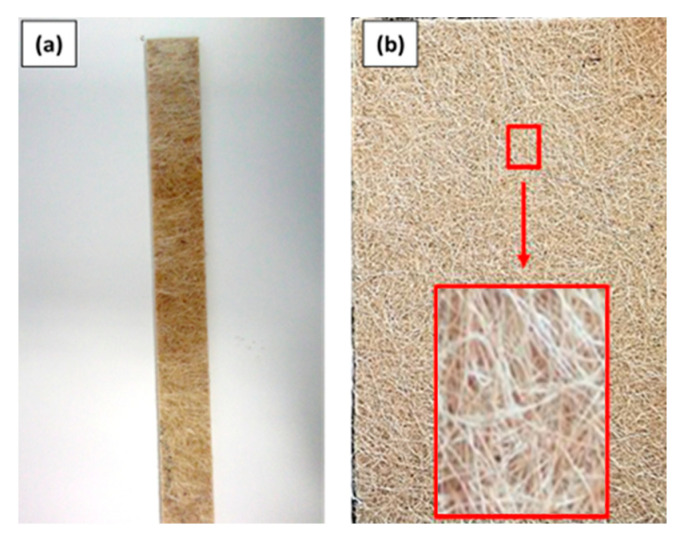
(**a**) Sample of sisal-mat composite post-processing, (**b**) Sisal Mat.

**Figure 2 polymers-16-00166-f002:**
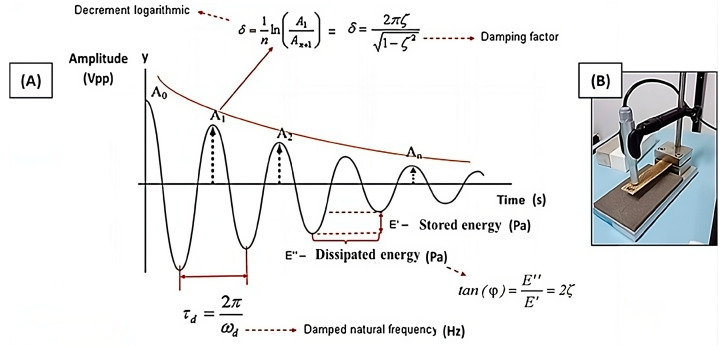
(**A**) Damped system response with exponential decay of the amplitude and (**B**) sample clamping configuration.

**Figure 3 polymers-16-00166-f003:**
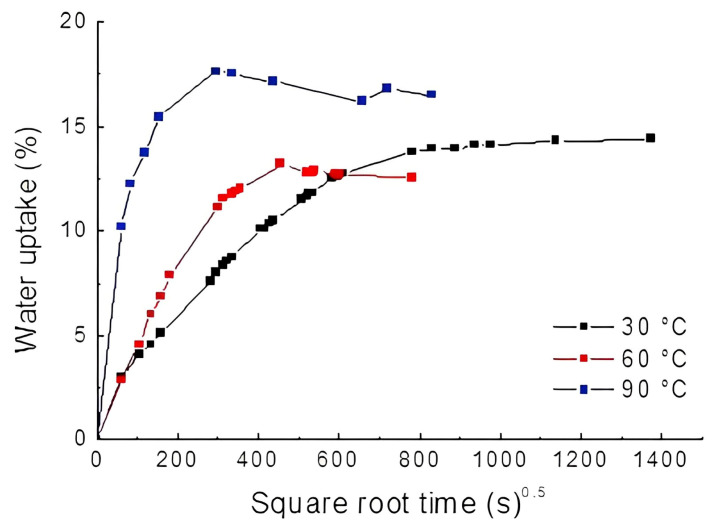
Water sorption curves for composites at 30 °C, 60 °C, and 90 °C.

**Figure 4 polymers-16-00166-f004:**
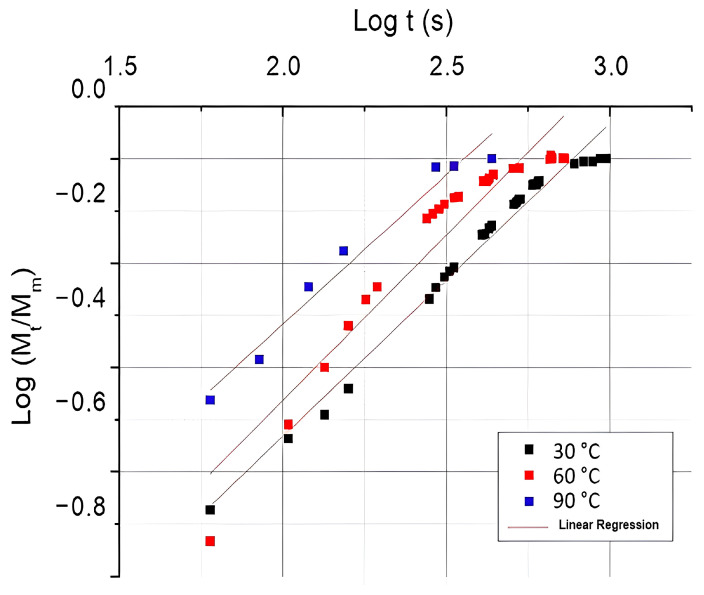
Linear regression model applied for composites at hydrothermal aging at 30 °C, 60 °C and 90 °C.

**Figure 5 polymers-16-00166-f005:**
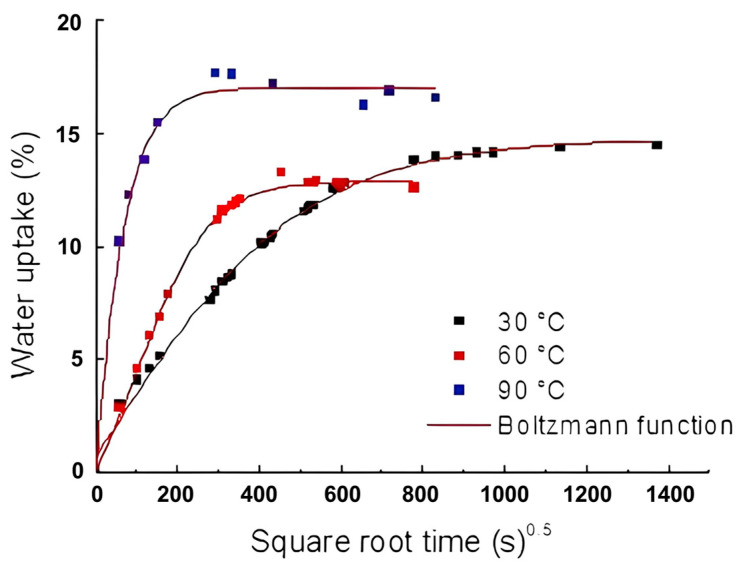
Boltzmann curve for sisal mat composites at different hydrothermal conditions.

**Figure 6 polymers-16-00166-f006:**
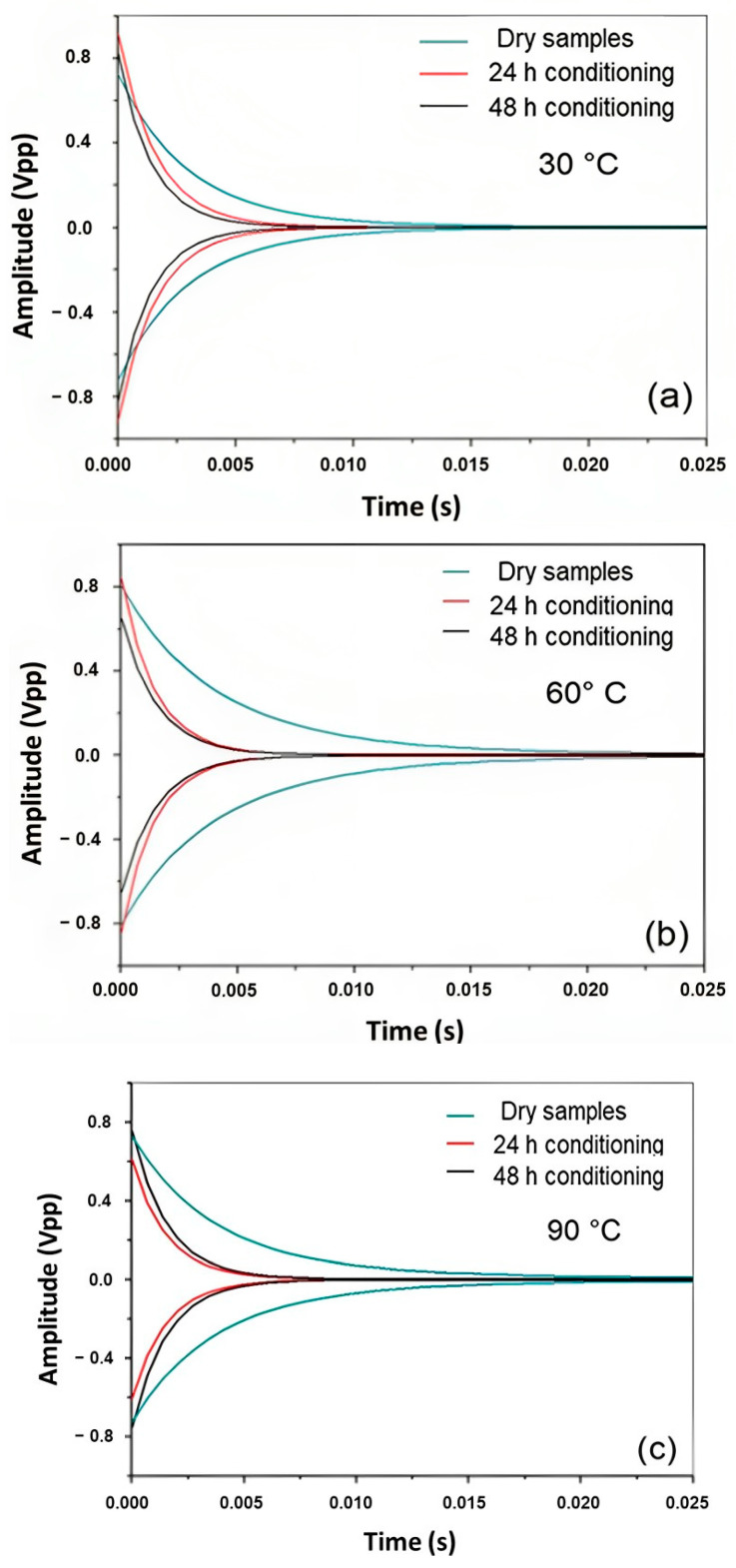
Damping at (**a**) 30 °C, (**b**) 60 °C and (**c**) 90 °C for the reference state and at 24 and 48 h of immersion.

**Figure 7 polymers-16-00166-f007:**
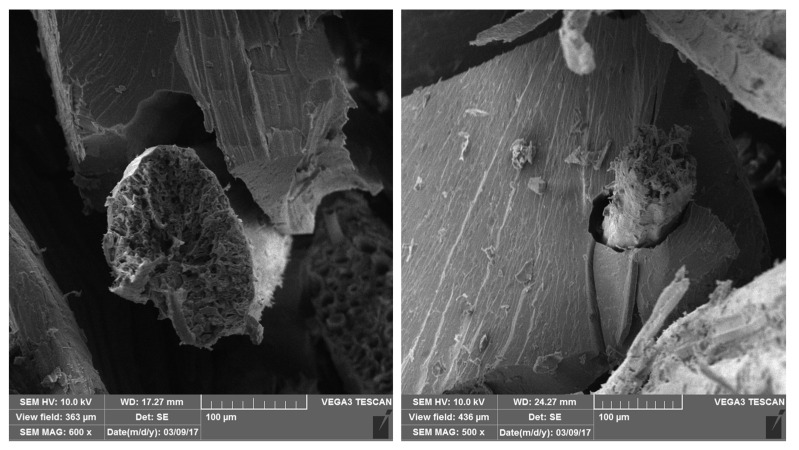
Scanning Electron Microscope—SEM (500×/600×): Microstructure damage by hydrothermal effects at 90 °C.

**Figure 8 polymers-16-00166-f008:**
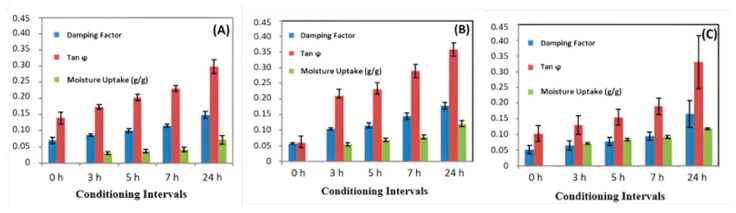
Water sorption (as moisture uptake in g/g) and damping parameters (damping factor and Tan φ)) for 30 °C (**A**), 60 °C (**B**) and 90 °C (**C**) hydrothermal conditions at short time intervals.

**Figure 9 polymers-16-00166-f009:**
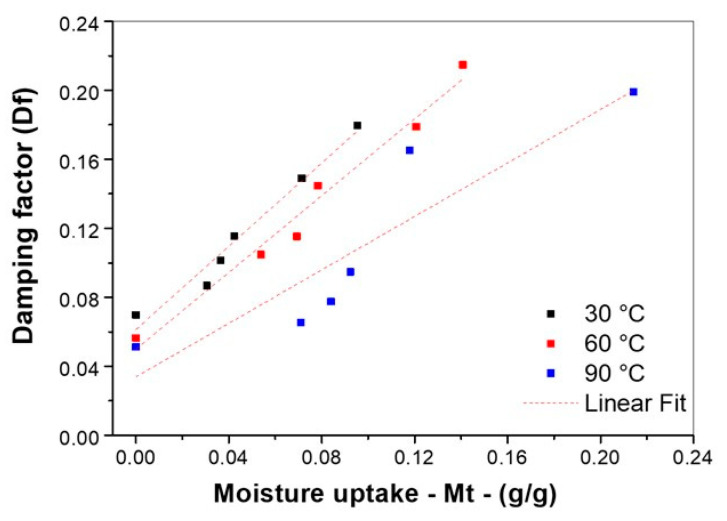
Linear correlation between moisture and damping factor.

**Table 1 polymers-16-00166-t001:** Equilibrium moisture for each conditioning temperature.

Temperature (°C)	Equilibrium Moisture (%)	Saturation Time (h)	Diffusion Coefficient (mm^2^·s^−1^)
30 ± 2	13.83 ± 0.29	264	6.01 · 10^−6^ ± 5.19 · 10^−7^
60 ± 2	12.85 ± 0.08	75	1.91 · 10^−5^ ± 7.16 · 10^−7^
90 ± 2	16.79 ± 0.28	31	1.27 · 10^−4^ ± 1.47 · 10^−5^

**Table 2 polymers-16-00166-t002:** Water sorption regression model constants for different environmental conditions.

Equation	log (*M_t_*/*M*_∞_) = log (*k*) + *n* log (*t*)		
Temperature	30 °C	60 °C	90 °C
R^2^	0.96	0.90	0.97
*n*	0.59 ± 0.017	0.63 ± 0.042	0.57 ± 0.045
*k*	0.0181	0.0185	0.035

**Table 3 polymers-16-00166-t003:** Water sorption regression model (non-linear Boltzmann sigmoid) constants for different environmental conditions.

Equation	*y* = *A*_2_ + (*A*_1_ − *A*_2_)/(1 + exp^((*x*−*x*_0_)/*dx*))^)		
Temperature	30 °C	60 °C	90 °C
Correlation (R^2^)	0.99	0.99	0.98
Initial (*A*_1_)	−12.97 ± 4.79	−3.78 ± 0.96	−105.13 ± 425.88
Final (*A*_2_) = *M*_∞_	14.75 ± 0.18	12.85 ± 0.08	16.98 ± 0.25
x_0_	6.63 ± 78.16	103.49 ± 11.32	−111.94 ± 281.97
Dx	246.38 ± 21.86	88.48 ± 5.71	61.36 ± 19.10

**Table 4 polymers-16-00166-t004:** Properties were obtained when carrying out the damping analysis for each hydrothermal condition over time.

T (°C)	Control Sample		24 h		48 h	
	Damping (ζ) × 10^−2^	Tan φ × 10^−2^	Damping (ζ) × 10^−2^	Tan φ × 10^−2^	Damping (ζ) × 10^−2^	Tan φ × 10^−2^
30	7 ± 0.9	14 ± 1.8	14.9 ± 1.0	29.8 ± 2.1	18 ± 1.3	35.9 ± 2.6
60	2.7 ± 0.9	5.4 ± 1.8	18.9 ± 2.6	37.8 ± 5.3	21.4 ± 1.9	42.9 ± 3.9
90	6.1 ± 0.7	12.3 ± 1.5	18.7 ± 1.2	37.4 ± 2.5	19.9 ± 0.9	39.8 ± 1.8

**Table 5 polymers-16-00166-t005:** Linear correlation model and quality of the fit for water sorption and damping factor.

Temperature	Equation	Correlation Coefficient (R^2^)
30 °C	*D*_f_ = 1.207 × *M_t_* + 0.0613	0.964
60 °C	*D*_f_ = 1.113 × *M_t_* + 0.0499	0.971
90 °C	*D*_f_ = 0.775 × *M_t_* + 0.0339	0.786

## Data Availability

Data are contained within the article, further details are available upon request from the corresponding author.

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
