# Peer review of "Effect of Hydrothermal Aging on Damping Properties in Sisal Mat-Reinforced Polyester Composites"

_polymers, 2024, doi:10.3390/polym16020166_

Round 1
Reviewer 1 Report
Comments and Suggestions for Authors
The manuscript titled "Effect of Hydrothermal Ageing on Damping Properties in Sisal Mats- Reinforced Polyester Composites" needs improvement in the following aspects.
1. Abstract has to be rewritten by including some details about the experiments and some salient results obtained.
2. Most of the articles (more than 50 %) used in the references are way too old. Authors may use only recently published articles (published during the last 5 years) for problem identification. Authors may refer to and include some of the following articles: https://www.mdpi.com/1420-3049/24/19/3538, https://doi.org/10.1007/978-981-16-8360-2_2.
3. Table 2 and 3 contains the values taken from references. It will be better if the authors verify them and include the measured values.
4. Authors have measured the values for 3, 5 and 7 hours also. But none of the graphs reflect these results. Kindly justify or include those values for better comparison.
5. Figures 2-4 have a poor readability. Authors may include figures in higher resolutions.
6. Morphological images have a very poor readability. Kindly include the full size images.
7. There are many language and grammatical errors all through the manuscript. It has to be proofread carefully.
Thanks in advance for your valuable time and support.
Comments on the Quality of English Language
Language needs significant revision
Author Response
Abstract has to be rewritten by including some details about the experiments and some salient results obtained.
Resp: We appreciate your feedback, and below you will find the changes we made, including details about the experiments and results."
Hydrothermal aging is a matter of considerable concern for natural fiber-reinforced polymers; it can alter dimensional stability and induce microcracks and macro strain on the composite structure. This study applied a sorption kinetic model and examined the effects of water on the damping factor of sisal mat-reinforced polyester composites. The experimental data were fitted well using a Boltzmann sigmoid function, suggesting a promising first step toward kinetic water sorption modeling. Additionally, a damping test was carried out using the impulse excitation technique, highlighting the composite material’s dynamic response under varying water absorption conditions. The result showed that damping exhibited sensitivity to water absorption, increasing significantly during the first 24 hours of immersion in water, then remained steady over time, inferring a critical time interval. An empirical model proved satisfactory with the correlation coefficient for sorption rates and damping of sisal mat polymeric composites.
Most of articles (more than 50%) used in the references are way too old. Authors may use only recently published articles (published during the last 5 years) for problem identification. Authors may refer to and include some of the following articles: https://www.mdpi.com/1420-3049/24/19/3538, https://doi.org/10.1007/978-981-16-8360-2_2
Resp: The requested references have been included and are presented below. Ramesh et al. (2022) has been replaced by Gudayu et al. (2022)
Gudayu A.D., Steuernagel L., Meiners D., Gideon R. (2022) “Effect of surface treatment on moisture absorption, thermal, and mechanical properties of sisal fiber.” Journal of Industrial Textiles. 2022;51(2_suppl):2853S-2873S. doi:10.1177/1528083720924774
Kandola, B.K., Mistik, S.I., Pornwannachai, W., Horrocks, A.R. (2021) “Effects of Water and Chemical Solutions Ageing on the Physical, Mechanical, Thermal and Flammability Properties of Natural Fibre-Reinforced Thermoplastic Composites.” Molecules, 26, 4581. https://doi.org/10.3390/ molecules26154581
Mazur, K., Kuciel, S. (2019). “Mechanical and Hydrothermal Aging Behaviour of Polyhydroxybutyrate-Co-Valerate (PHBV) Composites Reinforced by Natural Fibres.” Molecules, 24, 3538. doi:10.3390/molecules24193538.
Table 2 and 3 contains the values taken from references. It will better if the authors verify them and include the measured values.
Resp: The references provided in Tables 2 and 3 have been maintained and represent average ranges and other standard values of the physical-mechanical and chemical characteristics of sisal from the technical literature review article.
Authors have measured the values for 3, 5 and 7 hours also. But none of the graphs reflect these results. Kindly justify or include those values for better comparison.
Resp: This has been partially done.
Figures 2-4 have a poor readability. Authors may include figures in higher resolutions.
Resp: The changes have been made in accordance with the given instructions.
Morphological images have a very poor readability. Kindly include the full size images.
Resp: The images have been replaced with ones of higher quality and resolution.
There are many language and grammatical errors all through the manuscript. It has to be proofread carefully.
Resp: The article has been proofread and a certificate has been issued as part of the process.

Reviewer 2 Report
Comments and Suggestions for Authors
Summary: This study explores the moisture absorption and related damping properties of sisal fiber/polyester composites. Composites were manufactured using compression molding and then aged in water at three temperatures. A sigmoidal function describing the moisture uptake provided a good description of the results. Damping was found to scale mostly linearly with moisture uptake. The article was well written, and the results clearly communicated. However, the SEM analysis leaves a lot to be desired. I have provided several comments to the authors that must be resolved in a revised manuscript prior to publication. The work is interesting and valuable to the readers of this journal but requires major revisions before publication.
General Comments:
1. Section 2.1: Since sisal fiber mats are a somewhat unique composite reinforcement, it would be helpful to the readers if you provided a photo of the mats pre- and post-processing into composites.
2. Section 3.1: You reference equations (3) and (4), but neither are present in the manuscript.
3. Table 5: Why are there not error bars (standard deviation) on these results as in Table 4? I presume that you used the same data for each. This would help to validate that all values of n were consistently above 0.5 and that the apparent trends in k are meaningful.
4. Table 6: Please define all of the variables here. For example, what is dx? Is this an algebraic equation or an integral? It would be better to define equations in the text using standard formats. Also, why did the authors not report error bars here? Again, I assume that you used the same dataset as Table 4.
5. Figure 5: Please make all of the subfigures the same size. It is very difficult to compare them when the x-axes are not the same size.
6. Figure 6: What is the orange? It says polyester matrix, but it looks to be a representation that was drawn into the micrograph. This is not very convincing evidence. The SEM images to the left do not clearly show debonding. In particular, only focusing on one fiber lends itself to a lot of user discretion. More convincing images should be provided. Even better, the authors should consider providing a quantitative measurement of the debonding (either interface thickness or a relative percentage of exposed interface in images of the same dimensions that capture many distinct fibers).
7. Conclusions: The authors did not prove that matrix hydrolysis and plasticization occurred. There are several analytical tools that could provide this type of data (e.g., FTIR), but this claim is not substantiated by the current work and should not be included in the conclusions.
Specific Comments:
1. Section 2.3: Please include the interval at which the samples were weighed during hydrothermal aging.
2. Table 4: The final column should be titled “Diffusion Coefficient” or “Coefficient of Diffusion”. The same applies to the text.
3. Section 3.2, line 295: “microgafies” should be “micrographs”.
Comments on the Quality of English Language
Well-written and concise
Author Response
please find enclosed the certificate of english correction
answers reviewer2
Section 2.1 – Since sisal fibers mats are a somewhat unique composite reinforcement, it would be helpful to the readers if you provided a photo of the mats pre- and post-processing into composites.
Resp: Figures of the sisal mat and the post-processed composite have been added.
Section 3.1: You reference equations (3) and (4), but neither are present in the manuscript.
Resp: The requested suggestions have been duly corrected.
Table 5: Why are there not error bars (standard deviation) on these results as in Table 4? I presume that you used the same data for each. This would help to validate that all values of n were consistently above 0.5 and that the apparent trends in k are meaningful.
Resp: The values of” n” have been properly adjusted with the inclusion of the standard deviation.
Table 6: Please define all of the variables here. For example, what is dx? Is this an algebraic equation or an integral? It would be better to define equations in the text using standard formats. Also, why did the authors not report error bars here? Again, I assume that you used the same dataset as Table 4.
Resp: The corrections were made, and the Boltzmann equation with the definition of the involved parameters was introduced. Additionally, error bars were added.
Figure 6: What is orange? It says polyester matrix, but it looks to be a representation that was drawn into the micrograph. This is not very convincing evidence. The SEM images to the left do not clearly show debonding. In particular, only focusing on one fiber lends itself to a lot of user discretion. More convincing images should be provided. Even better, the authors should consider providing a quantitative measurement of the debonding (either interface thickness or a relativa percentage of exposed interface in images of the same dimensions that capture many distinct fibers).
Resp: The SEM images have been updated and reviewed.
Conclusions: The authors did not prove that matrix hydrolosys and plasticization occurred. There are several analytical tools that could provide this type of data(e.g FTIR), but this claim is not substantiated by the current work and should not be included in the conclusions.
Specific comments:
Section 2.3: Please include the interval at which the samples were weighed during hydrothermal aging.
Resp: The changes have been made.
Table 4: The final column should be titled “Diffusion Coefficient” or “Coefficient of Diffision”. The same applies to the text.
Section 3.2, line 295: “microgafies” should be “micrographs”
Resp: This has been corrected.

Reviewer 3 Report
Comments and Suggestions for Authors
In this work sisal fiber was used for preparation of polymer composite which was further characterized by DMA, SEM etc. and primarily damping behaviour was analyzed. Before possible publication, the following points should be addressed:
· Tensile properties were considered from reported literature which may vary from the actual. Moreover, merely tensile strength, Young’s modulus and density won’t give the real picture because here, in this work, it is much more related to further dynamic mechanical performances. Instead, linear density (tex or denier) and tenacity (g/tex or g/denier) should be reported which considers fiber length and shape irregularities too. The authors must carry out the tensile characteristics properly and correlate with the DMA result.
· For sisal, moisture regain is more important than merely mentioning water uptake. Moisture regain must be mentioned which could be calculated easily by environmental conditioning followed by oven drying.
· The images of sisal fiber and the composite should be included in the manuscript.
· What is meant by “Reference” in Table 7? Is it the control sample? If it is taken from any literature, mention the reference in the table, otherwise mention as “Control sample”.
· Scale bars are not clearly visible in SEM images.
Author Response
REVIEWER 3
Tensile properties were considered from reported literature, which may vary from the actual. Moreover, merely, tensile strength, Young’s modulus and density won’t give the real picture because here, in this work, it is much more related to further dynamic mechanical performances. Instead, linear density (tex or denier) and tenacity (g/tex or g/denier) should be reported which considers fiber length and shape irregularities too. The authors must carry out the tensile characteristics properly and correlate with the DMA result.
For sisal, moisture regain is more important than merely mentioning water uptake. Moisture regain must be mentioned which could be calculated easily by environmental conditioning followed by oven drying.
The images of sisal fiber and the composite should be included in the manuscript.
Resp: Figures of the sisal mat and the post-processed composite have been added.
What meant by “Reference” in Table 7? Is is the control sample? If it is taken from any literature, mention the reference in the table, otherwise mention as “Control sample”.
Resp: We have revised this accordingly.
Scale bars are not clearly visible in SEM images.
Resp: The SEM images have been updated and reviewed.

Round 2
Reviewer 1 Report
Comments and Suggestions for Authors
All the comments/queries raised by the reviewers were answered. The manuscript can be accepted in its current form. Best wishes to all the authors.
Author Response
REVIEWER 1 did not ask any changes.
Reviewer 2 Report
Comments and Suggestions for Authors
1The authors still have not proven that matrix hydrolysis and plasticization occurred. This should be removed from the conclusions or data supporting this conclusion should be acquired. Acquiring data that supports this conclusion would be preferred route, as it adds value to this work by showing the mechanism of property degradation.
Author Response
REVIEWER 2
“The authors still have not proven that matrix hydrolysis and plasticization occurred. This should be removed from the conclusions or data supporting this conclusion should be acquired. Acquiring data that supports this conclusion would be preferred route, as it adds value to this work by showing the mechanism of property degradation.”
Resp: We appreciate your feedback, and below you will find the changes we made, including details about the experiments and results."
The term “matrix hydrolysis and plasticization” was replaced by “matrix degradation”.
Reviewer 3 Report
Comments and Suggestions for Authors
The authors partially addressed the previous review comments:
Tensile properties were considered from reported literature, which may vary from the actual. Moreover, merely, tensile strength, Young’s modulus and density won’t give the real picture because here, in this work, it is much more related to further dynamic mechanical performances. Instead, linear density (tex or denier) and tenacity (g/tex or g/denier) should be reported which considers fiber length and shape irregularities too. The authors must carry out the tensile characteristics properly and correlate with the DMA result.
For sisal, moisture regain is more important than merely mentioning water uptake. Moisture regain must be mentioned which could be calculated easily by environmental conditioning followed by oven drying.
and these two comments were just skipped. The authors must address the comments and revise the manuscript before any possible publication.
Author Response
REVIEWER 3
Resp: We appreciate the revision suggestions and have outlined our response below.
Fiber properties were not the object of this study but as these characteristics may be useful to whom would use this paper, we reported the literature data. The reviewer is correct as they could vary from the actual fiber, this is only general ranges. To do not confound the reader, we decided to move this information to the supplementary material and reinforce that these are literature value range. As an random mantle was used the linear density or tenacity do not describe the mantle. The mechanical behavior of composites with large displacement and undergoing plastic strain is conditioned by the individual behavior of the materials and predicted by models such as misture rule. The dynamic behavior with small displacements and strain, within the elastic limits of the materials is sensitive to changes in internal friction and therefore to the damping observed in the vibration of the composite. The observation of the variation in the vibrational behavior of the composites is a methodological contribution of the work.
Studies with DMA can contribute to a better understanding of the effects of water adsorption on composites and we can certainly adopt them in future work. Sisal was not directly studied here instead the composite was, in this case moisture content is not easily calculated as suggested by the reviewer. Water uptake is more adequate in the present case.